Metagenomic identification of active methanogens and methanotrophs in serpentinite springs of the Voltri Massif, Italy

Brazelton William J. william.brazelton@utah.edu 1
Thornton Christopher N. 1
Hyer Alex 1
Twing Katrina I. 2
Longino August A. 1
Lang Susan Q. 3 4
Lilley Marvin D. 5
Früh-Green Gretchen L. 4
Schrenk Matthew O. 2
1 Department of Biology, University of Utah , Salt Lake City , UT , United States
2 Department of Earth and Environmental Sciences, Michigan State University , East Lansing , MI , United States
3 Department of Earth and Ocean Sciences, University of South Carolina , Columbia , SC , United States
4 Department of Earth Sciences, ETH Zurich , Zurich , Switzerland
5 School of Oceanography, University of Washington , Seattle , WA , United States
Kalyuzhnaya Marina
Electronic publication date: 2017 Jan 26
Publication date: 2017
Volume: 5
Electronic Location ID: e2945
Received 2016 Jul 22; Accepted 2016 Dec 27
Copyright: ©2017 Brazelton et al.
Copyright year: 2017
Copyright holder: Brazelton et al.
License: This is an open access article distributed under the terms of the Creative Commons Attribution License, which permits unrestricted use, distribution, reproduction and adaptation in any medium and for any purpose provided that it is properly attributed. For attribution, the original author(s), title, publication source (PeerJ) and either DOI or URL of the article must be cited.
License URL: https://creativecommons.org/licenses/by/4.0/

Keywords: Metagenomics, Serpentinization, Methanogenesis, Methanotrophy

Funding: NASA Astrobiology Institute Postdoctoral Fellowship University of Utah start-up funds Alfred P. Sloan Foundation’s Deep Carbon Observatory WJB received funding from a NASA Astrobiology Institute Postdoctoral Fellowship and University of Utah start-up funds. Additional funding to the Schrenk lab was provided by the Alfred P. Sloan Foundation’s Deep Carbon Observatory and the NASA Astrobiology Institute (NASA-CAN5 through the Carnegie Institution for Science). SQL and GLFG were funded by SNF Project 200020 14389. The funders had no role in study design, data collection and analysis, decision to publish, or preparation of the manuscript.

==============================
The production of hydrogen and methane by geochemical reactions associated with the serpentinization of ultramafic rocks can potentially support subsurface microbial ecosystems independent of the photosynthetic biosphere. Methanogenic and methanotrophic microorganisms are abundant in marine hydrothermal systems heavily influenced by serpentinization, but evidence for methane-cycling archaea and bacteria in continental serpentinite springs has been limited. This report provides metagenomic and experimental evidence for active methanogenesis and methanotrophy by microbial communities in serpentinite springs of the Voltri Massif, Italy. Methanogens belonging to family Methanobacteriaceae and methanotrophic bacteria belonging to family Methylococcaceae were heavily enriched in three ultrabasic springs (pH 12). Metagenomic data also suggest the potential for hydrogen oxidation, hydrogen production, carbon fixation, fermentation, and organic acid metabolism in the ultrabasic springs. The predicted metabolic capabilities are consistent with an active subsurface ecosystem supported by energy and carbon liberated by geochemical reactions within the serpentinite rocks of the Voltri Massif.

Introduction

Hydrothermal systems hosted in ultramafic rocks rich in serpentine minerals can support active and unique microbial communities (see Schrenk, Brazelton & Lang, 2013 for review). Microbial ecosystems associated with serpentinization (the process by which serpentine is formed) have been investigated by sampling seafloor hydrothermal chimneys (e.g., Kelley et al., 2005; Quéméneur et al., 2014), alkaline springs on land (Brazelton et al., 2013; Suzuki et al., 2013; Woycheese et al., 2015), and wells drilled into serpentinizing rocks (Tiago & Veríssimo, 2013; Cardace et al., 2013; Crespo-Medina et al., 2014). The microbial inhabitants of these environments are thought to feed on the hydrogen gas and organic compounds released as byproducts of serpentinization, but representative examples of specific organisms consuming specific products of serpentinization are lacking. For example, methane is typically abundant in serpentinite-hosted hydrothermal systems, but only a few studies have provided evidence for methanogenic or methanotrophic biological activity in such systems (Brazelton et al., 2011; Kohl et al., 2016).

The role of methane-cycling organisms in serpentinizing systems has been enigmatic since the discovery of thick biofilms of methanogens belonging to order Methanosarcinales in carbonate-brucite chimneys of the Lost City hydrothermal field (Schrenk et al., 2004). Whether these organisms are primarily involved in methane production or methane oxidation remains uncertain (Proskurowski et al., 2008; Bradley, Hayes & Summons, 2009; Brazelton et al., 2011; Lang et al., 2012; Méhay et al., 2013). Similar methanogens belonging to order Methanosarcinales have been identified in Prony Bay, New Caledonia, where freshwater serpentinizing fluids are venting into shallow marine waters and forming carbonate chimneys on the seafloor (Quéméneur et al., 2014). 16S rRNA genes associated with methanotrophic bacteria have been detected at Lost City (Brazelton et al., 2006) and Prony Bay (Quéméneur et al., 2014).

Recent studies of non-marine alkaline springs associated with serpentinization have identified 16S rRNA and/or methyl coenzyme-M reductase (mcrA) genes belonging to methanogens in California (The Cedars: Suzuki et al., 2013; Adobe Springs: Blank et al., 2009; Costa Rica: Sánchez-Murillo et al., 2014; and the Philippines: Woycheese et al., 2015). At the Tablelands Ophiolite (Newfoundland, CA), however, experimental incubations have been unable to detect methanogenesis at high pH (Morrill et al., 2014), and a metagenomic study at this site did not detect any methanogenesis genes (Brazelton, Nelson & Schrenk, 2012; Brazelton et al., 2013). 16S rRNA and mcrA sequences belonging to the ANME-1a group of anaerobic methanotrophic archaea, but not typical methanogens, were detected in serpentinizing groundwater in the Cabeço de Vide Aquifer in Portugal (Tiago & Veríssimo, 2013). Biogeochemical studies at The Cedars have inferred biological methanogenesis from isotopic signatures of methane and experimental results, but these studies have not included any biological characterizations of the responsible organisms (Morrill et al., 2013; Wang et al., 2015; Kohl et al., 2016). Thus, methanogens have been detected in some continental serpentinite springs, but previous reports have not provided any quantitative measurements of their environmental distributions nor any genomic or metagenomic sequences from methanogens. Evidence for methanotrophic bacteria in these systems is even more scarce: only a single study of non-marine alkaline springs associated with serpentinization has previously reported methanotrophic bacteria (Sánchez-Murillo et al., 2014).

In the Voltri Massif (Italy), ultrabasic springs (pH 11–12) rich in calcium and methane are issuing from serpentinites and rare lherzolites and are associated with the precipitation of large amounts of carbonate (Bruni et al., 2002; Cipolli et al., 2004; Boulart et al., 2013; Schwarzenbach et al., 2013). Inorganic carbon geochemistry of the springs and associated travertine deposits suggest autotrophic microbial activity in the subsurface (Schwarzenbach et al., 2013). A previous study has characterized the archaeal and bacterial communities within the travertine deposits formed when the springs reach the surface (Quéméneur et al., 2015), but there are no previous microbiological studies of the spring water. In this report we show that pH 12 springs at the Voltri Massif are transporting distinct archaeal and bacterial communities, including methanogenic archaea and methanotrophic bacteria, from subsurface habitats where they are likely to be supported by hydrogen gas, methane, and possibly other products of serpentinization-associated reactions. We provide insights into the biology and potential subsurface habitats of these organisms with metagenomic and experimental studies.

Materials and Methods

Location and sample collection

The ultrabasic springs that were investigated in this study (BR2: 44.4512°N, 8.7820°E; GOR34: 44.5970°N, 8.7833°E) are located in the Voltri Massif near Genoa, Italy (Fig. 1). Geochemical measurements of these springs were conducted in concert with the microbiological studies reported here and were reported by Schwarzenbach et al. (2013). At each of the BR2 and GOR34 locations, ultrabasic spring water and nearby river water were filtered through Millipore Sterivex 0.2 µm filter cartridges using a portable peristaltic pump. Measurements of pH and Eh were obtained with an Oakton PCS Testr 35 and an Oakton Testr 10, respectively, from pumped water that had passed through the peristaltic tubing but before the Sterivex filter (Table 1). Eh readings were corrected for the standard Ag/AgCl electrode (+200 mV).

Figure 1 Ultrabasic springs of the Voltri Massif in Italy were sampled at two locations.

(A) GOR34, including two springs and an adjacent river and (B) BR2, including one spring and an adjacent river. All water samples were collected by peristaltic pumping through a 0.2 µm filter cartridge. Photo credits: WJ Brazelton (A) and B Nelson (B).

Table 1 Chemical characteristics of ultrabasic springs and adjacent rivers of the Voltri Massif, Italy.

Sample name	pH	Eh (mV)	CH4 (µM)	H2(µM)	DIC (µM)	Sulfate (µM)	Sulfide (µM)	
BR2-spring-2012	12.1	−195	689	0.5	29	6.5	16.5	
BR2-spring-2013	12.3	−75	733	1.8	7.8	nm	16.2	
BR2-river-2012	8.1	450	bdl	0.2	3,031	130	bdl	
BR2-river-2013	8.0	311	bdl	bdl	2,678	nm	bdl	
GOR34-spring1-2011	12.2	−60	155	3.9	14	3.0	bdl	
GOR34-spring1-2013	12.3	−156	213	9.2	17.2	nm	18.6	
GOR34-spring3-2012	11.8	−202	201	26.8	29	bdl	17.7	
GOR34-river-2012	9.3	360	0.1	bdl	1,752	49	6.3	
GOR34-river-2013	9.5	240	bdl	bdl	1,670	nm	bdl	
Notes.

nm not measured

bdl below detection limit

GOR34-1-spring and GOR34-3-spring (Fig. 1A) were sampled from two shallow pools within a large deposit of travertine (Schwarzenbach et al., 2013). Each pool appeared to be fed by a point source of spring water at the bottom, into which peristaltic tubing was inserted as far as possible into the host rock (a few centimeters). The GOR34-1 pool was much smaller than that of GOR34-3, and it was emptied before sampling by siphoning water out of the pool. During pumping and sampling of water from the apparent subsurface source, the pump rate was adjusted to approximately match the rate of inflow, determined by a constant water level in the pool. The volume of water in the GOR34-3 pool did not visibly change during sampling, indicating that the maximum pump rate was significantly slower than the rate of spring water inflow into the pool. The pumped spring water from both GOR34-1 and GOR34-3 became more basic (higher pH) and more reducing (lower Eh) during pumping of the first few liters, after which the readings stabilized, and filtering commenced. GOR34-river was sampled from the surface of the adjacent river at a site of exposed, rapid flow ∼50 m upstream of the spring.

BR2-spring was sampled from a metal pipe (Fig. 1B) through which spring water was flowing at 492mLs−1 into the adjacent river (Schwarzenbach et al., 2013). After filtering >500 L of spring water from BR2-spring in 2012, less than 500 ng of DNA was recovered in initial extractions, which was reduced to 50 ng after pooling and final purification. Even lower yields were achieved in 2013 (Table 2). BR2-river was sampled from the surface of the river at a site of exposed, rapid flow ∼10 m upstream of the spring.

Table 2 Water samples collected from ultrabasic springs for metagenomic analyses.

Sample name	Sample date	Volume filtered (L)	ng DNA per L water	Cells per mLa	Bact:arch ratio	Bacterial 16S rDNA sequences	Archaeal 16S rDNA sequences	Metagenomic sequencesb	
BR2-spring-2012	30-Jun-2012	527	0.9	7.6 × 102	1.6	nm	nm	13,774,495	
BR2-spring-2013	22-Aug-2013	82	0.1	6.9 × 103	nm	nm	nm	8,047,152	
BR2-river-2012a	30-Jun-2012	1	136	2.4 × 104	nm	52,045	36,474	17,953,252	
BR2-river-2012b	30-Jun-2012	2.2	214	2.8 × 104	0.8	45,787	29,855	nm	
BR2-river-2013	22-Aug-2013	10	69	1.7 × 104	5	nm	nm	21,308,793	
GOR34-spring1-2011	18-Oct-2011	5	47	2.3 × 104	nm	15,056	724c	nm	
GOR34-spring1-2013a	24-Aug-2013	12	41	2.2 × 104	270	73,829	267c	nm	
GOR34-spring1-2013b	24-Aug-2013	35	4	1.4 × 104	10	107,493	101,232	27,882,181	
GOR34-spring3-2012	1-Jul-2012	24	<0.1	3.1 × 103	nm	nm	nm	14,967,449	
GOR34-river-2012a	1-Jul-2012	2	274	2.4 × 104	6,800	71,285	33,999	22,118,087	
GOR34-river-2012b	1-Jul-2012	2	294	7.1 × 104	0.7	79,012	115c	nm	
GOR34-river-2013a	24-Aug-2013	3.3	122	5.3 × 104	16	91,952	94,369	nm	
GOR34-river-2013b	24-Aug-2013	9.2	56	4.2 × 104	48	80,683	851c	36,276,889	
Notes.

nm not measured

a Cell concentrations are reported as the mean of three field replicates for each sample.

b Metagenomic sequences are reported as the number of quality-filtered read pairs.

c Samples with very low numbers of sequences were not included in any downstream analyses.

BR2-spring, BR2-river, and GOR34-river were sampled in June 2012 and August 2013. GOR34-1 was sampled in October 2011 and August 2013, while GOR34-3 was only sampled in June 2012. Field replicates were collected for all spring and river samples, but replicates were pooled post-DNA-extraction when necessary to accumulate enough DNA for sequencing. Furthermore, even after pooling of replicates, some samples contained only enough DNA for either 16S rRNA amplicon sequencing or shotgun metagenomic sequencing, but not both. For this reason, BR2-spring-2012, BR2-spring-2013, and GOR34-spring3-2012 are represented in this study by shotgun metagenomic sequences but not 16S rRNA amplicon sequencing surveys (Table 2). When available, field replicates are indicated with ‘a’ or ‘b’ at the end of the sample name, such as GOR34-river-2013a and GOR34-river-2013b.

Water chemistry

Water samples were collected for measurements of hydrogen, methane, sulfate, and sulfide. Gas samples were collected with 100 mL syringes by sampling 60 mL of water, minimizing any incorporation of air. Subsequently, 40 mL of nitrogen or helium gas was introduced to the syringe as headspace gas and the two phases were allowed to equilibrate. The gas phase was injected into evacuated vacutainers for later analysis. The concentrations of gas were determined by gas chromatography (GC), calibrated with commercially available gas standards of known concentration. For hydrogen, samples were injected in split/splitless mode (5:1 split) onto a CP-Molsieve 5 Å column (50 m × 0.32 µm, 30 µm thickness) at 175°C with a helium flow rate of 1.8 mL/min and detected with a Pulsed Discharge Detector. Methane was determined similarly but with a GS-Carbonplot column (30 m × 0.32 µm ID, 1.5 µm film thickness) held at 100°C for 3 min then ramped at 50 °C/min to 200 °C and held for 0.6 min. Detection was with a flame ionization detector. The detection limit was ∼0.6 nM for both hydrogen and methane.

Dissolved inorganic carbon (DIC) measurements were performed on samples that were collected by injecting 1–8 mL of 0.2 µm filtered water into Exetainers (Labco Limited, UK) that had been previously prepared with ∼100 µL of phosphoric acid and flushed with helium. Concentrations were determined by injecting aliquots of the headspace with a calibrated gas-tight syringe into the same GC set-up used for hydrogen. DIC concentrations from 2011 were reported by Schwarzenbach et al. (2013).

Sulfide concentrations were determined by the methylene blue method and detected at 610 nm on a spectrophotometer (Cline, 1969). Concentrations were compared to standard curves of freshly prepared sodium sulfide. The detection limit was 3.1 µM. For sulfate, 100 mL of water was collected in high density polypropylene bottles and stored cool until analysis. Concentrations were determined by a DX-120 ion chromatograph equipped with an IonPac As14 column (4 × 250 mm). The detection limit was 1.0 µM. Organic acids were analyzed by high performance liquid chromatography (HPLC) by the method of Albert & Martens (1997) with minor modifications. After derivatization with 2-nitrophenyl hydrazine and 1-ethyl-3-(3-dimethylaminopropyl) carbodimide hydrochloride, acids were separated on a Prevail Organic Acid C18 column and detected at 400 nm. Adipic acid was used as an internal standard.

Enumeration of microbial cells

Fluids (50 mL per field replicate; three replicates per sample in Table 2) were preserved in the field for cell abundance enumeration at a final concentration of 3.7% formaldehyde and stored at 4°C. In the laboratory, preserved fluids (5–20 ml each) were filtered through a 0.2 µm black polycarbonate filter, and captured cells were stained with DAPI and counted with an epifluorescence microscope. At least 30 fields were counted for each field replicate and used to calculate the average cell concentration per mL of fluid for that replicate sample. The numbers reported in Table 2 reflect the mean cell concentration for three field replicates from each sample.

DNA extraction and sequencing

Filters were placed on ice immediately and stored within a few hours at liquid nitrogen temperature in a vapor shipper (MVE SC4/2V) for frozen transport to the home laboratory. Each Sterivex filter was extracted according to protocols modified from those described in Huber, Butterfield & Baross (2002) and Sogin et al. (2006). Briefly, extractions were performed by lysis via freeze/thaw cycles and lysozyme/Proteinase K treatment and purified with phenol-chloroform extractions and precipitation in ethanol. Environmental DNA quantification was performed with the Qubit fluorometer (ThermoFisher). Extraction blanks never yielded quantifiable DNA and were not sequenced. Further purification of DNA was conducted with QiaAmp (Qiagen) columns according to the manufacturer’s instructions for purification of genomic DNA.

Purified DNA was submitted to the Josephine Bay Paul Center, Marine Biological Laboratory (MBL) at Woods Hole for amplicon sequencing of archaeal and bacterial 16S rRNA genes with an Illumina MiSeq platform. The V4–V5 hypervariable region of the bacterial 16S rRNA gene was targeted with one forward (518F; CCAGCAGCYGCGGTAAN) and three reverse primers (926R; CCGTCAATTCNTTTRAGT; CCGTCAATTTCTTTGAGT; CCGTCTATTCCTTTGANT). The V4–V5 hypervariable region of the archaeal 16S rRNA gene was targeted by a combination of five forward primer variants (517F; GCCTAAAGCATCCGTAGC; GCCTAAARCGTYCGTAGC; GTCTAAAGGGTCYGTAGC; GCTTAAAGNGTYCGTAGC; GTCTAAARCGYYCGTAGC) and a single reverse primer (958R; CCGGCGTTGANTCCAATT). Additional methodological details and description of primer development are published in Nelson et al. (2014) and Topçuoğlu et al., (2016). Amplicon sequences were screened for quality, including chimera-checking with UCHIME (Edgar et al., 2011), by the MBL as previously described (Huse et al., 2014a), and high-quality merged sequences were published on the Visualization and Analysis of Microbial Population Structures (VAMPS) website (Huse et al., 2014b).

The MBL also conducted shotgun metagenomic sequencing of a subset of these samples (see Table 2). Metagenomic libraries were constructed with the Nugen Ultralow Ovation kit according to the manufacturer’s instructions. Paired-end sequencing with a 100 cycle Illumina HiSeq run generated partial ∼30 bp overlaps, and six libraries were multiplexed per lane. 16S rRNA amplicon sequences are publicly available at the VAMPS database (http://vamps.mbl.edu) under the project code DCO_BRZ and sample code Serp_LIG. Shotgun metagenomic data is publicly available in MG-RAST under IDs 4545477.3, 4545478.3, 4545479.3, 4545480.3, 4537863.3, 4537864.3, 4537868.3, and 4537869.3. All sequence data related to this study are also available via the SRA identifier SRP049438 and BioProject PRJNA265986.

Quantitative PCR of archaeal and bacterial 16S rRNA gene copies

The 16S rRNA gene copies of Bacteria and Archaea were quantified via quantitative PCR on a BioRad CFX Connect Optics Module with the BioRad SsoAdvanced SybrGreen assay and domain-specific primers targeting the V6 region of the 16S rRNA gene. Primers 958F and 1048R were used for Archaea, and primers 967F and 1064R were used for Bacteria (previously published in Sogin et al. (2006) and described on the VAMPS website https://vamps.mbl.edu/resources/primers.php). Gene copy numbers were calculated by plotting quantification values from environmental samples onto standard curves generated by amplification of DNA from Methanocaldococcus jannaschii (for Archaea) and Escherichia coli (for Bacteria) with the domain-specific primers. Bacteria:Archaea ratios were then calculated with gene copy numbers normalized to sample size, i.e., the volume of fluid filtered for that sample.

Analysis of 16S rRNA amplicon data

Additional quality screening of 16S rRNA amplicon sequences from VAMPS was conducted with the mothur (v.1.34.2) software platform (Schloss et al., 2009) to remove sequences with >9 homopolymers and >0 ambiguous bases. This screening step removed only 124 bacterial sequences and 5 archaeal sequences. The sequences were then pre-clustered with the mothur command pre.cluster (diffs = 1), which reduced the number of unique sequences from 410,517 to 322,879 for bacteria and from 138,629 to 95,867 for archaea. This pre-clustering step removes rare sequences most likely created by sequencing errors (Schloss, Gevers & Westcott, 2011). The final unique sequence types were considered to be the operational taxonomic units (OTUs) for this study. We chose not to cluster sequences any more broadly (e.g., 97% sequence similarity) because additional clustering inevitably results in a loss of biological information without reliably improving sequencing error and because no arbitrary sequence similarity threshold can be demonstrated to correspond to consistent species-like units across all taxa. Taxonomic classification of all OTUs was performed with mothur using the SILVA reference alignment (SSURefv119) and taxonomy outline (Pruesse, Peplies & Glöckner, 2012). Differences in the relative abundances of OTUs between groups of samples were measured with the aid of the R package edgeR v.3.6.8 (Robinson, McCarthy & Smyth, 2010) as recommended by McMurdie & Holmes (2014). Results were visualized with the aid of the R package phyloseq v.1.9.13 (McMurdie & Holmes, 2013).

Analysis of metagenomic data

Detailed documentation of all computational analyses in this study, including actual software commands, are provided on the Brazelton lab’s website (https://baas-becking.biology.utah.edu/data/category/18-protocols), and all custom software and scripts are available at https://github.com/Brazelton-Lab. Quality-filtering of shotgun metagenomic sequences included identification and removal of artifactual sequences with cutadapt v.1.9 (Martin, 2011) as follows. Reads found to have Illumina adapters starting at their 5′-end were discarded, and reads containing Illumina adapters towards the 3′-end were trimmed where the first adapter began. Identical and 5′-prefix replicates were also removed, as suggested by Gomez-Alvarez, Teal & Schmidt (2009). Nucleotides (0–2) at the beginning and end of reads were also cropped from all reads in that sample if those positions exhibited nucleotide frequencies inconsistent with the nucleotide frequency distribution for the rest of the read. Low-quality bases were removed from the ends of the reads, and the remaining sequence was scanned 20 base pairs at a time and trimmed where the mean quality score fell below a score of 8. Reads that did not pass a minimum length threshold of 62 bp after quality and adapter trimming were removed from the dataset. Phylogenetic affiliations of the unassembled, quality-checked paired reads were determined with PhyloSift v.1.0.1 (Darling et al., 2014).

Each metagenomic dataset was processed individually as described above. The four metagenomes from ultrabasic springs and the four metagenomes from adjacent rivers were then combined for two pooled assemblies with Ray Meta v.2.3.1 (Boisvert et al., 2012). A kmer of 41 was chosen after manual inspection of assemblies with kmer values of 31, 41, 51, and 61. High-quality reads from each sample were mapped onto the pooled assembly with Bowtie2 v.2.2.6 (Langmead & Salzberg, 2012) to obtain sample-specific coverages. The Prokka pipeline (Seemann, 2014) was used for gene prediction and functional annotation. The arguments –metagenome and –proteins were used with Prokka v.1.12 to indicate that genes should be predicted with the implementation of Prodigal v.2.6.2 (Hyatt et al., 2010) optimized for metagenomes and then searched preferentially against a custom protein database. The database provided was the Kyoto Encyclopedia of Genes and Genomes FTP release 2016-09-26 (Ogata et al., 1999). Abundances of metabolic pathways were obtained by mapping KEGG protein IDs and their normalized counts calculated with HTSeq v.0.6.1 (Anders, Pyl & Huber, 2015) onto the FOAM ontology (Prestat et al., 2014) with MinPath (Ye & Doak, 2009) as implemented in HUMAnN2 v.0.6.0 (Abubucker et al., 2012). Similar results were also achieved by annotating both pooled assemblies with the UniProtKB protein database (Consortium, 2015) and MetaCyc pathways (Caspi et al., 2014).

Contigs from the pooled spring assembly were binned according to their tetranucleotide frequencies with the aid of an emergent self-organized map (ESOM) as implemented by Databionic ESOM tools v.1.1 (Ultsch & Mörchen, 2005) and as described by Dick et al. (2009). The resulting ESOM was annotated with phylogenetic markers classified by PhyloSift and selected predicted protein functions of interest identified by KEGG ID. Completeness and contamination of ESOM bins were evaluated with CheckM v.1.0.4 (Parks et al., 2015). Contamination levels as measured by CheckM do not always agree with the percentage of minority taxa reported by PhyloSift because CheckM only identifies contamination when single-copy markers are present as multiple, divergent sequences and because PhyloSift results are abundance-weighted.

Metabolic activity assays

During each sampling expedition, water samples collected from each of the BR2 and GOR34 springs were incubated with one of several 13C-labeled carbon sources (obtained from Cambridge Isotope Laboratories, Tewksbury, MA): bicarbonate (NaH13CO3), formate (Na13CHOO), acetate-13C1 (NaCH313COO), acetate-13C2 (Na13CH3COO), propionate (NaCH3CH213COO), or methane (13CH4). Each 99% 13C-labeled carbon compound was diluted 1:100 with the corresponding non-13C-labeled compound to produce a final δ13C of +691‰ , and this compound was present in each incubation at a final concentration of 0.1 M after addition of 3 mL of water sample. No buffering nor additional amendments were applied to the incubations, and the experiments are assumed to have proceeded at the in situ pH of the spring water. ‘Dead’ control incubations were conducted by filter-sterilizing the spring water through a 0.2 µm syringe filter in the field and prior to incubation. Three live replicates and three dead replicates from each spring were incubated for each treatment. Methanogenesis experiments received an addition of 5 cc H2 gas to each replicate. Methanotrophy experiments received an addition of 5 cc 13CH4 that had been diluted 1:10 with non-13C-labeled methane. All incubations were conducted in gas-tight glass Exetainers (Labco Limited, UK) flushed with N2 and 2–5% H2 in a Coy anoxic chamber and flushed again outside the chamber and slightly over-pressured for transport with N2 gas. Oxic methanotrophy experiments were conducted after exposing the contents to room air by opening the lids for a few minutes prior to adding 13CH4. Some methanogenesis experiments were treated with a reducing agent in the form of 0.3 mL of 1% dithiothreitol (DTT) or 2.5% sodium sulfide, but no differences were observed with or without the additional reducing agent. After incubation at ambient temperature (20–35 °C) in the dark for 4–8 days, Exetainers were injected with 0.2 mL of 50% phosphoric acid to convert all dissolved inorganic carbon into CO2, and the 13C/12C ratios of CO2 and CH4 were measured by gas chromatography isotope ratio mass spectrometry using a Thermo Fisher Delta V Plus Isotope Ratio Mass Spectrometer, interfaced with a Trace gas chromatograph, a GC IsoLink interface and a ConFlo IV at the Stable Isotope Laboratory of the ETH in Zurich.

Results

Biogeochemical contrasts between ultrabasic springs and surface rivers

At both the BR2 and GOR34 sites, the spring water had much higher pH, lower Eh (i.e., more reducing), fewer particulates, and lower biomass than the adjacent river (Tables 1 and 2). The high pH (up to 12.3) and low Eh are consistent with the spring water being sourced from ultramafic rocks where serpentinization-like reactions are occurring now or have occurred in the past (Alt et al., 2013). The difference in particulates was noticeable due to the much greater volumes of spring water, compared to surface river water, that could be filtered through a single Sterivex cartridge before clogging (more than 20–50 L for springs, only 2–5 L for surface rivers). The difference in biomass was also evident from the DNA yield: the GOR34 springs yielded 3–5-fold less DNA per liter of water compared to the adjacent river, and the BR2 spring yielded >1,000-fold less DNA per liter of water compared to the adjacent river. The cell densities (as measured by enumeration of DAPI-stained cells under a microscope) of the BR2 spring (102–103 cells per mL of water) were 2–40-fold lower than the cell densities of the adjacent river (Table 2). The cell densities of the GOR34 springs were 2–20-fold lower than the adjacent river.

BR2 spring waters contained higher methane and lower hydrogen concentrations compared to the GOR34 springs (Table 1). At both sites, spring water contained much lower dissolved inorganic carbon (DIC) than in the adjacent rivers, likely due to the precipitation of calcium carbonate at high pH (Schwarzenbach et al., 2013; Alt et al., 2013). Sulfate was also very low in all springs, though detectable at BR2. Sulfide was elevated in all springs compared to the adjacent rivers.

Taxonomic survey of bacterial 16S rRNA genes

The bacterial taxonomic composition of the pH 12 spring GOR34-spring1 was distinct from that of surface rivers, as measured by high-coverage Illumina MiSeq sequencing of bacterial 16S rRNA gene amplicons. Comamonadaceae (a member of order Burkholderiales, class Betaproteobacteria) was the most abundant family in GOR34-spring1 in 2013 (35–40% of all bacterial 16S rRNA amplicon sequences, File S1). The most abundant OTU within this family is 100% identical to the most abundant OTU recovered from ultrabasic serpentinite springs at the Tablelands Ophiolite in Newfoundland, Canada (Brazelton et al., 2013) and to the most abundant OTU in borehole fluids from the Coast Range Ophiolite Microbial Observatory in California (Crespo-Medina et al., 2014). This same bacterial taxon has been isolated from serpentinite springs at The Cedars (northern California) by Suzuki et al. (2014), who have proposed the novel genus Serpentinomonas for these organisms. Curiously, Comamonadaceae sequences only comprised 14% of all bacterial sequences in GOR34-spring1 in 2011. Instead, the most abundant sequences were those classified as Candidate Division OD1 (21% of sequences in GOR34-spring1 in 2011 but only 4% in 2013).

Comamonadaceae sequences were also well represented in surface river samples (File S1), but the Comamonadaceae OTUs from river samples were distinct from the Comamonadaceae OTUs from ultrabasic spring samples (only 94% sequence identity over 373 bases between the most abundant Comamonadaceae OTU in GOR34-spring1-2013b compared to the most abundant Comamonadaceae OTU in GOR34-river-2013). Neither of these sequences could be classified at a lower taxonomic rank than family. This result highlighted the need to conduct comparisons of bacterial community compositions at the level of individual OTUs (File S2), rather than taxonomic classifications.

Differential abundance of bacteria

Although the bacterial community compositions of ultrabasic springs and the adjacent rivers were generally distinct, the most abundant OTUs in rivers were also present, at low relative abundances, in ultrabasic springs, and the converse was also true. In other words, the most abundant 16S rRNA OTUs at each site were not completely exclusive to that site. For example, the most abundant OTU in GOR34-river-2013b (classified as Sphingomonadaceae) occurred 8,515 times in that sample and 0 and 4 times in GOR34-spring1-2013a and GOR34-spring1-2013b, respectively. Conversely, the most abundant OTU in GOR34-spring1-2013b (the Comamonadaceae Serpentinomonas OTU described above) occurred 32,866 times in that sample and 71 and 0 times in GOR34-river-2013a and GOR34-river-2013b, respectively (File S2). As with any environmental study, these sequences could have appeared in multiple locations because of natural mixing in the environment or due to accidental contamination during sampling or sequencing. Regardless of the cause, the source of shared sequences can be inferred by making the parsimonious assumption that the site of higher abundance is closer to that organism’s true habitat. Therefore, we looked for potential subsurface-specific organisms by testing for significant differences in the abundances of individual OTUs between ultrabasic spring water and the adjacent river.

For this analysis, we contrasted the counts of OTUs in GOR34-spring1 to those in GOR34-river. Each site was represented by two field replicates collected in 2013 (Table 2). In Fig. 2, each data point of the plot represents the difference in abundance of a single OTU between GOR34-spring1 and GOR34-river. The plot’s x-axis displays the mean abundance (in units of log2 counts per million) of each OTU across all four samples (two GOR34-spring1 replicates and two GOR34-river replicates). Red data points indicate OTUs whose differential abundance between GOR34-spring1 and GOR34-river passed a significance test (false discovery rate < 0.05) implemented by the edgeR package (Robinson, McCarthy & Smyth, 2010). Spring-enriched OTUs include the Comamonadaceae described above as well as a group of uncultured Bacteroidetes known as aquatic group ML635J-40, Candidate Division OD1, an uncultured group within the Thermoanaerobacterales, and Methylococcaceae (Fig. 2A). Sequences that were not consistently enriched in one sample type compared to the other are represented by black points in Fig. 2. These sequences could represent potential contaminants or truly cosmopolitan species, or their abundances may be correlated to factors not captured by the study design. The data visualization approach in Fig. 2 highlights the key differences between the spring and river microbial communities by acknowledging the reality that all environmental samples are mixtures formed by multiple sources of organisms and without making the naive and extreme assumption that all shared sequences must be some kind of contamination.

Figure 2 Bacterial and archaeal sequences enriched in ultrabasic springs.

Taxonomic classifications of bacterial (A) and archaeal (B) operational taxonomic units (OTUs, defined as unique 16S rRNA sequences) that were identified as significantly enriched in the ultrabasic serpentinite spring GOR34-spring1 compared to the adjacent river. Each data point in the plots represents one OTU’s mean abundance across all samples in the analysis and its differential abundance in the comparison GOR34-spring1 versus GOR34-river. Mean abundance is reported as log2 counts per million; a value of 16 corresponds to 23,000 sequences for that OTU. Red data points represent OTUs with differential abundances that are significantly enriched (false discovery rate < 0.05) in either GOR34-spring1 or GOR34-river. The relative abundances of these significantly enriched OTUs are reported as a percentage of all significantly enriched sequences and grouped into taxonomic classifications at the family level in the colored bar charts. Sections of the bar chart were labeled with the corresponding family where possible, and the abbreviations are defined by bold font in the right-hand legends.

Taxonomic survey of archaeal 16S rRNA genes

As with the bacterial communities, the archaeal communities in the ultrabasic spring waters were distinct from those in the surface rivers, as measured by high-coverage Illumina MiSeq sequencing of archaeal 16S rRNA gene amplicons. More than 88% of total archaeal sequences in GOR34-spring1-2013, for example, belong to the Methanobacteriaceae family of methanogens. The closest match (∼97% sequence similarity) in the GenBank, VAMPS, and SILVA databases to these Methanobacteriaceae sequences is a clone from subsurface water collected in a deep South Africa mine (GenBank DQ230925). The taxonomic composition of GOR34-river was nearly completely different; ∼72% of sequences in both 2012 and 2013 were assigned to Marine Group I, within the Thaumarchaeota (File S3). As with the bacterial sequences, the most abundant archaeal OTUs in GOR34-river were also present, at low relative abundances, in GOR34-spring1, and the converse was also true. For example, the most abundant OTU in GOR34-spring1-2013b occurs 89,407 times in that sample, 127 times in GOR34-river-2013, and 24 times in GOR34-river-2012 (File S4).

We identified potential subsurface-specific archaeal 16S rRNA OTUs by contrasting the archaeal sequences from GOR34-spring1 to those of GOR34-river. Unfortunately, GOR34-spring1-2013b was the only spring sample to yield sufficient DNA and sufficient read coverage for this analysis of archaeal 16S rRNA sequences. GOR34-river is represented in this analysis by two samples: GOR34-river-2012a and GOR34-river-2013a. Not surprisingly, considering the differential abundances reported above, the Methanobacteriaceae sequences were identified as the most abundant OTUs that were significantly enriched in GOR34-spring1 (Fig. 2B). Other families of methanogens in addition to Methanobacteriaceae were identified in the 16S rRNA sequences, but none were significantly more abundant in the ultrabasic springs compared to the adjacent river. One sequence belonging to family Methanosaetaceae (order Methanosarcinales) was enriched in the GOR34 spring compared to the adjacent river in 2013, but Methanosaetaceae sequences were equally abundant in the river in 2012 (Files S3–S4). The only OTUs that could be considered significantly enriched in GOR34-river belong to the Thaumarchaeota Marine Group I family. The variability of less abundant OTUs between the two samples from GOR34-river prevented them from passing our significance test, even though many of them were extremely rare or absent in GOR34-spring1.

Taxonomic survey of shotgun metagenomic sequences

Phylogenetic classifications (from PhyloSift) of unassembled metagenomic sequences were largely consistent with the 16S rRNA amplicon results described above. For example, even though Comamonadaceae was the most common family identified in metagenomic sequences from the ultrabasic springs as well as the river at GOR34 (File S5), the sample types differed at the genus level: Hydrogenophaga was the most common member of Comamonadaceae in the GOR34-spring1 and GOR34-spring3 metagenomes, whereas Acidovorax was the predominant representative of Comamonadaceae in GOR34-river (File S6). Comamonadaceae sequences were also abundant in metagenomic data from the ultrabasic spring at BR2, although this site appears to be dominated by Desulfovibrionales, a member of Deltaproteobacteria (File S5).

Metagenomic sequences predicted to represent methanogens (family Methanobacteriaceae) comprised ∼1% of shotgun sequences in BR2-spring-2012 and GOR34-spring3-2012 (File S5). Curiously, no methanogen sequences were found in GOR34-spring1-2013b (File S6), despite their dominance in the archaeal 16S rRNA amplicon data from this same sample (Fig. 2). None of the 28 million sequence pairs in this metagenome were classified as domain Archaea, so it is possible that the ratio of bacterial to archaeal DNA in this sample was very high. A quantitative PCR assay of bacterial and archaeal 16S rRNA gene copies indicated a 10:1 ratio of Bacteria:Archaea (Table 2) in GOR34-spring1-2013b, which would predict a minority, but perhaps not a complete absence, of archaeal sequences in its metagenome. The Bacteria:Archaea ratio for BR2-spring-2012 was less than 2:1, and only 3% of shotgun metagenomic sequences from this sample were classified as Archaea, indicating a lack of direct correspondence between quantitative PCR results and numbers of shotgun sequences. When archaea were detected in metagenomic sequences from the rivers adjacent to the BR2 and GOR34 springs, they were dominated by Thaumarchaeota (File S5), consistent with the archaeal 16S rRNA amplicon results (Fig. 2). No Methanobacteriaceae sequences were detected in the metagenomic data from either river.

Metagenomic assembly and metabolic pathway prediction

Shotgun metagenomic sequences were assembled into contiguous genomic fragments (contigs), which were used for predictions of genes, protein functions, and metabolic pathways. Because the four metagenomes from ultrabasic springs (BR2-spring-2012, BR2-spring-2013, GOR34-spring1-2013b, and GOR34-spring3-2012) were similar to each other in taxonomic composition (Files S5–S6), they were combined in a pooled assembly called ‘combined-spring’. The N50 (length of median contig) of this pooled assembly was 2.4 kb (Table 3), and the longest contig was 99 kb. Of the 104,651 predicted proteins in the combined-spring assembly, 49,794 could be annotated with the KEGG database. A large proportion (45%) of predicted proteins had no matches in any of the protein databases and could not be assigned a function (Table 3). The four metagenomes from adjacent rivers (BR2-river-2012a, BR2-river-2013, GOR34-river-2012a, and GOR34-river-2013b) were also combined for a pooled assembly called ‘combined-river’, which has similar assembly statistics to the combined-spring assembly (Table 3).

Table 3 Assembly and annotation statistics for the pooled metagenomic assemblies and the five ESOM bins.

	Combined -spring	Combined -river	ESOM Bin 1	ESOM Bin 2	ESOM Bin 3	ESOM Bin 4	ESOM Bin 5	
# Contigs (≥500 bp)	55,894	40,355	228	157	123	169	88	
# Contigs (≥25,000 bp)	122	186	7	0	11	9	6	
Total length (≥500 bp)	89,619,646	73,320,758	1,261,593	1,129,542	1,117,180	1,686,028	1,124,653	
Total length (≥25,000 bp)	4,783,632	9,953,584	330,655	0	342,191	277,131	191,679	
Largest contig (bp)	99,214	304,939	93,351	24,214	37,024	41,177	34,915	
N50 (length of median contig)	2,367	3,515	4,908	9,513	12,857	15,069	16,218	
Predicted genes	108,150	91,693	1,280	974	820	1,453	965	
Predicted protein-encoding genes	104,651	89,606	1,256	963	805	1,423	933	
Predicted rRNA-encoding genes	77	67	0	0	0	0	0	
Predicted tRNA-encoding genes	2,267	1,349	16	11	12	20	27	
Predicted other RNA-encoding genes	1,155	671	8	0	3	10	5	
Predicted proteins annotated with KEGGa	71,856	67,170	1,014	874	714	1,271	855	
Predicted proteins annotated with UniprotKB	34	19	0	0	0	0	0	
Predicted proteins annotated with Pfam	482	367	4	7	2	3	3	
Predicted proteins annotated with HAMAP	58	30	0	0	0	0	0	
Predicted proteins annotated with CLUSTERS	212	151	1	0	0	1	0	
Predicted proteins with no annotation	32,009	21,869	237	82	89	148	75	
Notes.

a Predicted proteins were first mapped to the KEGG database, and any remaining proteins without a predicted function were then mapped to the other databases.

The abundance of metabolic pathways was determined by mapping predicted protein functions (as annotated with KEGG) and their sequencing read coverages onto the FOAM ontology. Many of the most abundant predicted pathways in the ultrabasic springs involved transporters and stress responses, possibly indicating cellular responses to the extreme environmental conditions. The most abundant energy-conserving metabolic pathways included those involved in fermentation, methanogenesis, sulfur oxidation, denitrification, methylotrophy, and hydrogen oxidation and production (File S7). The presence of genes associated with the aerobic oxidation of hydrogen and methanogenesis are expected from the dominance of the Hydrogenophaga and Methanobacteriaceae sequences described above. The abundance of several pathways involved in fermentation and hydrogen production is consistent with the presence of Clostridiaceae and Bacteroidetes, and the detection of methylotrophy pathways is consistent with the presence of Methylococcaceae in the ultrabasic springs (File S7).

Methanogenesis pathways using carbon dioxide and formate were among the most abundant pathways in the spring metagenomes, but other FOAM-defined methanogenesis pathways predicted to utilize acetate and methylated compounds were also predicted to be present in these springs (File S7). To better understand the evidence for different kinds of methanogenesis pathways in these springs, we examined the presence of each key methanogenesis-associated gene. All of the steps in the ‘core’ methanogenesis pathway were represented by at least one predicted protein (as defined by KEGG) in the combined-spring assembly (green highlighting in Fig. 3). The only exception is the lack of sequences encoding the enzyme Hmd (H2-forming methylenetetrahydromethanopterin dehydrogenase), but this gene is not present in many methanogens (Fricke et al., 2006), including the five Methanobacterium genomes currently available in the Integrated Microbial Genomes database (img.jgi.doe.gov).

Figure 3 Predicted methanogenesis proteins in the combined-spring metagenomic assembly.

Diagram of methanogenesis pathways from carbon dioxide, formate, acetate, methanol, and methylamines with associated protein homologs identified with KEGG IDs. Green-highlighted proteins are predicted to occur in the combined-spring metagenomic assembly, and yellow-highlighted proteins are predicted to occur in both the combined-spring and combined-river assemblies. The black border indicates proteins identified in the Methanobacteriaceae bin (ESOM Bin 1 in Fig. 4 and Table 5). The diagram is modified from Ferry (2010).

All of the genes encoding formate dehydrogenase (Fdh), acetate kinase (Ack), phosphotransacetylase (Pta), AMP-forming acetyl-CoA synthetase (Acs), and carbon monoxide dehydrogenase/acetyl-CoA decarbonylase/synthase (Cdh/ACDS) are present in the combined-spring assembly. The presence of these genes indicates the genetic potential for both formate and acetate to be used as substrates for methanogenesis, although these proteins can be utilized by non-methanogens as well, which can explain why some of these proteins are also found in the combined-river assembly (yellow highlighting in Fig. 3). Two of the three genes encoding Mta (methanol-specific methylcobalamin:coenzyme M methyltransferase) are also present, suggesting methanol as another potential substrate for methanogenesis.

Binning of metagenomic assembly

In order to examine the distribution of predicted protein functions and metabolic pathways among putative organisms within the combined-spring assembly, fragments of metagenomic contigs were binned according to their tetranucleotide compositions with an emergent self-organizing map (ESOM). Each point in the ESOM in Fig. 4 represents a contig or a ∼5 kb fragment of a contig if the original contig was larger than 5 kb. In Fig. 4A, contigs containing PhyloSift taxonomic markers that correspond to the enriched taxa in Fig. 2 are colored accordingly. Figure 4B is the same map as Fig. 4A, but in this case, fragments are color-coded according to the presence of predicted proteins associated with metabolic functions of interest (File S8). Five bins of metagenomic fragments with relatively homogenous taxonomic assignments were identified in this ESOM (Table 4). Unfortunately, none of the five bins contained any 16S rRNA genes, so direct links to the 16S rRNA gene amplicon data could not be made, and comparisons with PhyloSift taxonomic classifications of protein-encoding taxonomic markers are reported below instead.

Figure 4 An emergent self-organizing map (ESOM) constructed with the combined-spring metagenomic assembly.

Each point represents a fragment of metagenomic assembly, and points are arranged in space according to the similarities of their tetranucleotide compositions. Dark gray areas are valleys where highly similar sequences cluster together. Light areas of the map are high topographic ridges that separate dissimilar sequences. The same ESOM is shown here twice: metagenomic fragments are color-coded in (A) to indicate PhyloSift classifications of taxonomic markers and in (B) to show which fragments are predicted to encode protein functions of particular interest to this study. Based on this information, five bins of sequences were selected from the map for further evaluation (Table 5).

Table 4 Characteristics of ESOM bins identified in Fig. 4.

	Completeness	Contamination	Strain heterogeneity	Taxonomy	Metabolic potential	
Bin 1	44.11%	1.2%	75%	Methanobacteriaceae	Methanogenesis; nitrogen fixation	
Bin 2	42.16%	11.72%	3.23%	Comamonadaceae/ Xanthomonadaceae	Carbon fixation (Rubisco); organic carbon degradation	
Bin 3	27.59%	1.72%	0%	Bacteroidetes	Hydrogen production; fermentation	
Bin 4	47.98%	6.24%	34.88%	Methylococcaceae	Methylotrophy	
Bin 5	60.34%	6.03%	100%	Desulfovibrionales	Hydrogen oxidation; dissimilatory sulfate reduction	

ESOM Bin 1 (44% complete, 1% contamination) corresponds to the Methanobacteriaceae 16S rRNA sequences that were enriched in the ultrabasic springs (Fig. 2) because 88% of its sequences encoding taxonomic markers were classified as Methanobacteriaceae by PhyloSift (File S6). Furthermore, genomes belonging to genus Methanobacterium were the most common source of best hits when sequences belonging to ESOM Bin 1 were competitively recruited to a set of all methanogen genomes available from GenBank. ESOM Bin 1 sequences were recruited to many other types of methanogens as well, indicating that this bin does not have a single nearly identical whole-genome match in GenBank. ESOM Bin 1 includes many of the methanogenesis-associated sequences in the combined-spring assembly (black borders in Fig. 3), including genes involved in the utilization of formate, acetate, and methanol as substrates for methanogenesis. Two of the genes encoding nitrogenase (nifDK) were also identified in this bin, suggesting the potential for nitrogen fixation (File S8), which is consistent with the low levels of fixed nitrogen expected to be in the spring water (Cipolli et al., 2004). Curiously, ESOM Bin 1 does not include any of the genes that encode formylmethanofuran dehydrogenase, which catalyzes the first step of methanogenesis when carbon dioxide is the carbon source (Fig. 3). Although the absence of genes in an incomplete metagenomic bin cannot be definitive, it is nevertheless interesting to observe that the only methanogens known to lack all of the formylmethanofuran dehydrogenase genes (according to a comparison of KEGG annotations on JGI’s Integrated Microbial Genomes website) are members of genus Methanomassiliicoccus, which use methanol as their carbon source and cannot grow on carbon dioxide.

ESOM Bin 2 (42% complete, 12% contamination) includes sequences that were classified as Comamonadaceae (43% of taxonomic markers) and Xanthomonadaceae (27% of taxonomic markers) (File S6). Both of these bacterial families were also among the most abundant taxa identified in the bacterial 16S rRNA data (Fig. 2). Repeated attempts to partition this bin to separate the Comamonadaceae and Xanthomonadaceae sequences were unsuccessful, suggesting shared sequence compositions and/or mis-assembly. ESOM Bin 2 encodes RuBisCo (ribulose-1,5-bisphosphate carboxylase/oxygenase), the key enzyme in the Calvin-Benson-Bassham pathway of carbon fixation. The predicted protein sequence of the large subunit has 83% identity and 91% similarity with the Comamonadaceae RuBisCo sequence identified in metagenomic sequences from a serpentinite-hosted ultrabasic spring at the Tablelands Ophiolite in Newfoundland, Canada (Brazelton, Nelson & Schrenk, 2012). Nearly all predicted proteins in the combined-spring assembly involved in the degradation of chlorinated aromatic compounds are also found in ESOM Bin 2 (File S7), and these sequences have close matches in other Comamonadaceae genomes. This bin does not include any sequences encoding a NiFe-hydrogenase, but there is at least one large contig in the combined-spring assembly that encodes a NiFe-hydrogenase with 84% amino acid identity with the group 1 NiFe-hydrogenase from Serpentinomonas strain H1, a Comamonadaceae isolate from a serpentinite spring at The Cedars, California (Suzuki et al., 2014).

ESOM Bin 3 (28% complete, 2% contamination) is a collection of sequences with PhyloSift classifications consistent with that of the Bacteroidetes uncultured aquatic group ‘ML635J-40’, which was identified as the second-most abundant bacterial taxon in the ultrabasic springs (Fig. 2). 92% of all taxonomic markers in ESOM Bin 3 were classified as Bacteroidetes (File S6), although these sequences were somewhat evenly distributed among several different families within the Bacteroidetes, consistent with the undetermined phylogenetic placement of the ML635J-40 aquatic group (Nolla-Ardèvol, Strous & Tegetmeyer, 2015). This bin includes 50% of all combined-spring metagenomic sequences assigned to the FOAM pathway ‘Pyruvate fermentation to acetate III’ and is also rich in sugar transporters (File S7). ESOM Bin 3 encodes multiple [FeFe]-hydrogenases with high similarity to those encoded by other Bacteroidetes including Lentimicrobium saccharophilum (GenBank GAP44922.1) and Alistipes sp. ZOR0009 (GenBank WP_047449271).

ESOM Bin 4 (48% complete, 6% contamination) corresponds to the Methylococcaceae 16S rRNA sequences identified in Fig. 2. 60% of taxonomic markers in this bin were classified as family Methylococcaceae, and 82% were classified as order Methylococcales (File S6). Bacteroidetes sequences comprised 12% of taxonomic markers in this bin. Predicted protein sequences for particulate and soluble methane monooxygenase (pmoCAB, K10944, K10945, K10946; mmoXYBZDC, K16157, K16158, K16159, K16160, K16161, K16162) were prominent in ESOM Bin 4 (File S8), and they shared 83–90% amino acid identity with sequences from Methylobacter and Methylomicrobium species. The most abundant Methylococcaceae 16S rRNA gene amplicon sequences in the ultrabasic springs are most similar to a clone recovered from a deep mine in South Africa (Blanco et al., 2014), which is most closely related to Methylosoma difficile (Rahalkar, Bussmann & Schink, 2007).

ESOM Bin 5 (60% complete, 6% contamination) appears to capture much of the genomic content associated with the Desulfovibrionales 16S rRNA sequences that dominate the BR2-spring-2012 metagenome. 83% of taxonomic markers in this bin were classified as Desulfovibrionales (File S6). Furthermore, [NiFe]-hydrogenase sequences in ESOM Bin 5 have high similarity to other Desulfobivrionales genomes; for example, one predicted protein sequence has 90% identity with the hydrogenase 2 large subunit of Desulfonatronum lacustre (Pikuta et al., 2003). The Desulfonatronum genus includes several alkaliphilic, hydrogen-oxidizing species, although there are no reports of Desulfonatronum growth above pH 10. In addition to [NiFe]-hydrogenase (hydAB; K18008, K00437), most of the combined-spring metagenomic sequence coverage of carbon monoxide dehydrogenase (cooS; K00198), acetyl-coA synthase (acsB; K14128), three subunits of acetyl-CoA decarbonylase/synthase complex (Cdh/ACDS; K00194, K00197, K00198), dissimilatory sulfite reductase (dsrAB; K11180 –K11181), heterodisulfide reductase (hdrABC; K03388, K03389, K03390), and phosphate transacetylase (pta; K13788) were included in this bin (File S8).

Metabolic activity assays

The activity of methanogenesis, methanotrophy, and oxidation of organic acids in ultrabasic springs was tested by measuring the conversion of 13C-labeled carbon compounds during incubations of spring water samples at ambient temperatures (∼20–25 °C) in the dark. A single plus symbol in Table 5 indicates that the 13C ratio of the product for that reaction was higher in all three live replicates compared to all three dead replicates, which had been filter-sterilized prior to incubation. Double plus symbols indicate that the live replicates were 10–60‰ heavier than the corresponding dead replicates, and triple plus symbols indicate that the live replicates were 100–700‰ heavier than the dead replicates.

Table 5 Detection of metabolic activities in incubation experiments with spring water.

	BR2-spring -2011	BR2-spring -2012	GOR34-spring1 -2010	GOR34-spring1 -2012	GOR34-spring3 -2011	GOR34-spring3 -2012	
CO2→ CH4	–	–	–	–	nm	–	
Formate → CH4	–	–	–	–	nm	–	
Acetate-C1→ CH4	–	–	–	–	+ +  +	–	
Acetate-C2→ CH4	–	–	–	–	–	–	
Acetate-C1→ CO2	–	nm	+ +	nm	+ +	nm	
Acetate-C2→ CO2	–	nm	+	nm	–	nm	
Propionate → CO2	+	nm	–	nm	+	nm	
CH4→ CO2	+ +	–	+ +  +	+ + (+)a	+ +	–	
Notes.

nm not measured

a Third (+) symbol indicates stronger signal when exposed to oxygen.

Methanogenesis from acetate was detected in GOR34-spring3-2011. Unfortunately, insufficient material was collected from this spring in 2011 for high-quality sequencing studies, and subsequent metabolic activity experiments failed to detect methane production, most likely due to technical difficulties with performing these experiments. Curiously, methane production was only detected from the carboxyl group (C1 atom) of acetate. Acetoclastic methanogenesis generates methane by reduction of the methyl group (C2 atom), so this result suggests that the carboxyl group of acetate was first oxidized to inorganic carbon (carbon dioxide, bicarbonate, or carbonate) and then utilized by methanogens. Indeed, oxidation of acetate to carbon dioxide was observed in GOR34-spring3-2011. Methane oxidation to carbon dioxide was also observed at least once in all three springs. Methane oxidation signals were greatly increased when incubations were exposed to oxygen (indicated by the third plus symbol in GOR34-spring1-2012 in Table 5) and were completely eliminated by addition of a reducing agent. This result strongly indicates that the measured methane oxidation was aerobic, although some activity was also detected even without oxygen exposure.

Discussion

Methanogenesis

This study of ultrabasic springs at the Voltri Massif provides the first metagenomic evidence for methanogens and methanogenesis pathways in a continental serpentinizing system. All genes required for methanogenesis are present in the combined-spring assembly (Fig. 3), and many of these were collected in a single metagenomic bin corresponding to the Methanobacteriaceae taxa that are enriched in the ultrabasic springs (Fig. 2). The combined-spring assembly includes the genetic potential for the utilization of carbon dioxide, formate, acetate, and methanol as methanogenic substrates. Some of the genes involved in carbon dioxide, formate, and acetate utilization were also found in genomic bins for non-methanogens and in the adjacent rivers, but the two proteins required for the utilization of methanol as a methanogenic substrate were unique to the Methanobacteriaceae bin. Furthermore, the Methanobacteriaceae bin lacked all known genes for the utilization of carbon dioxide for methanogenesis. There are no previous examples of organisms within family Methanobacteriaceae that can use methanol as a substrate, so this result should be confirmed with additional genomic and experimental data. Nevertheless, methanol and formate are among the single carbon compounds predicted to be abiogenic products of serpentinization (Shock, 1992; Shock & Schulte, 1998; Seewald, Zolotov & McCollom, 2006), so these metagenomic data are consistent with the possibility that Methanobacteriaceae methanogens at this site are directly supported by subsurface serpentinization reactions.

We detected active methanogenesis in GOR34-spring3 (Table 5), which is assumed to have occurred at close to the in situ pH of the spring (pH 11.8). To our knowledge, this would represent the highest pH known to support methanogenesis, although no pH measurements were made during or after the incubation experiments. Methane produced during this experiment was derived from the carboxyl group (C1 atom) of acetate, and no methane was produced from the methyl group (C2 atom). This is contrary to the expectation for acetate-based methanogenesis, which produces methane by reduction of the methyl group. Furthermore, Methanobacteriaceae methanogens are not known to be able to use acetate as a substrate for methanogenesis. Therefore, it is more likely that the carboxyl group was oxidized to inorganic carbon (activity for which was also detected in this spring; see Table 5) and subsequently utilized for methanogenesis directly or indirectly. Unfortunately, methanogenesis from inorganic carbon (carbon dioxide, bicarbonate, or carbonate) was not measured during that experiment, and subsequent experiments failed to detect methanogenesis from any substrates. These results are also not easily reconciled with the absence of genes required for the utilization of carbon dioxide for methanogenesis in the Methanobacteriaceae ESOM bin. Thus, the genomic evidence for the potential of formate, acetate, and methanol to support methanogenesis by Methanobacteriaceae organisms remains only incompletely confirmed by metabolic activity experiments. Future experimental studies designed to test the utilization of specific carbon compounds during active methanogenesis at high pH would provide stronger evidence for the link between subsurface carbon sources and methanogens in serpentinite springs.

Methanobacteriaceae (as well as orders Methanosarcinales and Methanocellales) have been previously detected in travertine deposits formed by ultrabasic springs of the Voltri Massif nearby, but not the same as, the springs described here (Quéméneur et al., 2015). The presence of Methanobacteriaceae in the carbonate deposits raises the question of whether they primarily inhabit surface or subsurface environments. Methanobacteriaceae and predicted proteins in the methanogenesis pathway are abundant in the BR2 spring (Files S5–S8), which is not submerged by an overlying pool, does not appear to come into contact with surface travertine deposits, and can be sampled directly from the spring’s source (Fig. 1). Therefore, the high abundance of Methanobacteriaceae in this spring suggests a subsurface habitat for these methanogens, although it is possible that these organisms could be active in both subsurface springs and surface deposits where biofilms may create anoxic micro-environments.

Hydrogen metabolism

In addition to hydrogen gas consumption by methanogens, the Voltri Massif springs encode abundant genes for hydrogen oxidation (File S8). [NiFe]-hydrogenases associated with Comamonadaceae and Desulfovibrionales, which dominate the bacterial communities of the ultrabasic springs, were abundant in the metagenomic data from these springs. The Comamonadaceae 16S rRNA, [NiFe]-hydrogenase, and Rubisco sequences are highly similar to those reported for Comamonadaceae from other continental sites of serpentinization including the Cabeço de Vide Aquifer (Tiago & Veríssimo, 2013), the Tablelands Ophiolite (Brazelton, Nelson & Schrenk, 2012), The Cedars (Suzuki et al., 2014), and the Coast Range Ophiolite Microbial Observatory (Crespo-Medina et al., 2014). The presence of these organisms in alkaline springs appears to be a consistent indicator of serpentinization (Schrenk, Brazelton & Lang, 2013), and the genus Serpentinomonas has been proposed for them (Suzuki et al., 2014). Serpentinomonas 16S rRNA sequences have also been recovered from surface carbonate deposits formed by a spring at the Voltri Massif near to those reported in this study (Quéméneur et al., 2015), which is consistent with the interpretation that these organisms live at the surface or the shallow subsurface where serpentinizing fluids mix with the oxygenated atmosphere (Schrenk, Brazelton & Lang, 2013; Brazelton et al., 2013).

By contrast, the Desulfovibrionales represented by ESOM Bin 5 appear to be anaerobic, hydrogen-utilizing, sulfate-reducing bacteria and may live in anoxic, deep subsurface habitats underlying the springs. These bacteria, like the methanogens described above, were most abundant in the BR2 spring, which also contains higher sulfide levels. Therefore, this spring appears to provide more direct access to organisms that may have been recently active in anoxic, methanogenic, and sulfate-reducing subsurface habitats.

[FeFe]-hydrogenase sequences were also abundant, and many were associated with the unclassified Bacteroidetes group enriched in the ultrabasic springs (Fig. 2). These genes are typically involved in fermentative hydrogen production and appear to have a broad taxonomic distribution in our study, as illustrated by their wide distribution in the ESOM (Fig. 4B). Clostridiales, which were also enriched in the springs, are expected to encode some of the [FeFe]-hydrogenase sequences (e.g., Mei et al., 2014), but no ESOM bins with taxonomic markers consistently classified as Clostridiales could be identified.

These results are consistent with previous observations of abundant hydrogenases in metagenomic data from marine and continental serpentinite springs (Brazelton, Nelson & Schrenk, 2012). Phylogenetic analyses of those sequences indicated that [NiFe]-hydrogenases associated with Comamonadaceae and [FeFe]-hydrogenases associated with Firmicutes are likely to be involved in hydrogen consumption and production, respectively. The co-occurrence of Betaproteobacteria (often order Burkholderiales, including family Comamonadaceae) and Firmicutes (in particular order Clostridiales) is a common feature of serpentinite springs and perhaps the mixing of deep subsurface and surface fluids in general (reviewed by Schrenk, Brazelton & Lang, 2013; see also Purkamo et al., 2016). This report contributes to that emerging trend by providing another example of Comamonadaceae organisms with [NiFe]-hydrogenases and also reports [NiFe]-hydrogenases in Desulfovibrionales and [FeFe]-hydrogenases in Bacteroidetes in serpentinite springs for the first time.

Methanotrophy

Aerobic methane oxidation was detected in all three of the ultrabasic springs in this study (Table 5), and the Methylococcaceae family of aerobic methanotrophs was among the most enriched taxa in the ultrabasic springs (Fig. 2). The genomic bin corresponding to this family (ESOM Bin 4) includes sequences encoding both the particulate and soluble forms of methane monooxygenase (pmoABC and mmoXYBZDC) (Ward et al., 2004). Being aerobic, these bacteria are unlikely to be active in anoxic subsurface environments. However, a completely surface-exposed habitat also seems unlikely because they were not detected in a travertine deposit formed by one of these springs (Quéméneur et al., 2015). Therefore, they may inhabit a shallow subsurface mixing zone where low levels of oxygen reach methane-rich subsurface water, similar to the inferred habitat for the aerobic, hydrogen-oxidizing Comamonadaceae described above.

Fermentation

Multiple fermentation pathways were enriched in the ultrabasic springs, and fermentation genes were widely distributed among metagenomic fragments in the ESOM (Fig. 4). Candidate Division OD1 (now called Parcubacteria) was enriched in the ultrabasic springs here and has also been found in serpentinite springs at The Cedars that are expected to represent deeper, subsurface habitats (Suzuki et al., 2013). These bacteria are expected to be involved with fermentation (Hu et al., 2016), but we were unable to verify this due to the lack of any ESOM bins with a strong representation of Candidate Division OD1. Additional work will be required to better characterize the genomes of these organisms, which have been reported to be small and lacking genes typically thought to be essential for basic cellular processes (Nelson & Stegen, 2015; Brown et al., 2015).

An uncultured group of Bacteroidetes known as ‘ML635J-40 aquatic group’ was also enriched in the ultrabasic springs and was well-represented in the ESOM. The bin of metagenomic sequences associated with this group (ESOM Bin 3) encoded proteins predicted to be involved in pyruvate fermentation, sugar uptake, and hydrogen production. A member of ML635J-40 aquatic group was recently identified in a pH 10 anaerobic reactor in which it was predicted to be primarily responsible for hydrolysis of organic matter and supply of hydrogen gas to methanogens (Nolla-Ardèvol, Strous & Tegetmeyer, 2015). The Bacteroidetes in the BR2 and GOR34 ultrabasic springs could have a similar syntrophic relationship with methanogens, although the source of hydrolyzable organic matter in this case is unclear. The ML635J-40 aquatic group was not detected in a previous study of surface travertine deposits (Quéméneur et al., 2015), indicating that they do not inhabit a completely surface-exposed environment. Although great care was taken to avoid surface contamination during field sampling, including measures to sample the subsurface source of the spring rather than the overlying pool (see ‘Methods’), it remains possible that the Bacteroidetes sequences in the GOR34 springs represent pool-dwelling organisms that are dependent on terrestrial organic matter that falls into the surface-exposed pools. This interpretation is consistent with the lower abundance of Bacteroidetes in the BR2 spring (File S6), which is not submerged by an overlying pool (Fig. 1). It is also possible that they inhabit a shallow subsurface transition zone, similar to that inferred above for the Comamonadaceae and Methylococcaceae. Future studies should investigate the carbon source for these organisms in order to determine whether they are supported by compounds synthesized by subsurface serpentinization-associated reactions.

Formate and acetate metabolism

Both formate and acetate are present at elevated concentrations in fluids venting from the chimneys of the Lost City hydrothermal field (Lang et al., 2010), and formate is expected to be an abiogenic product of serpentinization-associated reactions (Shock, 1992; Shock & Schulte, 1998; McCollom & Seewald, 2001; McCollom & Seewald, 2003; Seewald, Zolotov & McCollom, 2006). Metagenomic sequences predicted to encode proteins associated with formate and acetate metabolism were abundant (File S8) and widespread among taxa (green and blue points, respectively, in Fig. 4B) in the combined-spring assembly, including the ESOM bins representing Methylococcaceae and Desulfovibrionales (Table 4). The wide taxonomic distribution of these genes (formate c-acetyltransferase, formate dehydrogenase, acetate kinase, phosphotransacetylase, AMP-forming acetyl-CoA synthetase, and carbon monoxide dehydrogenase/acetyl-CoA decarbonylase/synthase (Cdh/ACDS)) clearly indicates that they are not restricted to methanogens in this system and that this ecosystem features a broad metabolic potential for utilizing formate and acetate. We measured organic acid concentrations in Voltri Massif springs in 2010, before the microbiological studies described here. Formate and acetate were present at low but detectable concentrations (0.4 and 2.7 µM, respectively) at GOR34-spring3 and were below detection in all other springs. It is unclear whether the low levels indicate a lack of production or rapid consumption, but our initial metabolic activity assays have demonstrated that acetate can be oxidized to carbon dioxide at in situ conditions characterized by extremely high pH. Additional work is required to test whether the availability of formate and acetate are correlated with the activity of specific genes, including those identified by this study. Furthermore, future studies should attempt to demonstrate in the case of each group of organisms reported here whether formate and acetate are consumed or produced, whether they are cycled intracellularly, and whether they are obtained from or released into the environment.

Conclusions

The potential significance of serpentinization-supported microbial ecosystems has been widely recognized since the discovery of the Lost City hydrothermal field (Kelley et al., 2001), but we are still in the early stages of characterizing the organisms and pathways that may benefit from the subsurface geochemical reactions associated with serpentinization. This study of ultrabasic springs at the Voltri Massif, Italy provides the first evidence for active methanogenesis and aerobic methanotrophy in continental serpentinite springs as well as genomic information about the specific organisms likely to be responsible for these processes. Both methanogenesis and methanotrophy were active at very high pH, potentially raising the upper pH limit known to support both processes. These potential activity experiments were not intended to quantify in situ rates, however, and do not preclude the possibility that the vast majority of methane in these springs is produced by non-biological processes. The data reported here also contribute additional metagenomic evidence for the importance of hydrogenases involved in both the consumption and production of hydrogen gas in serpentinite systems.

The unavailability of inorganic carbon at the extremely high pH of these springs is likely to be a limiting factor for autotrophic activity, and small organic compounds such as formate, acetate, and methanol may serve as the primary carbon sources in such ecosystems. The absence of genes in the Methanobacteriaceae bin required for the reduction of carbon dioxide may be a reflection of the lack of inorganic carbon in this system. Furthermore, we found evidence for inorganic carbon as a substrate for methanogenesis only after it had been liberated by the oxidation of acetate. The genetic potential for metabolism of formate and acetate is widespread among taxa in this environment, which is consistent with an ecosystem being at least partially supported by organic carbon synthesized by subsurface serpentinization-associated reactions.

These genomic and metabolic data were obtained despite the extremely low biomass of these springs, suggesting that our results are the first clues into the few organisms capable of survival, and potentially growth, in the harsh conditions of the pH 12 springs. Because many of these organisms were only found in the spring waters and not in surface travertine deposits (Quéméneur et al., 2015), they are likely to represent inhabitants of subsurface environments in the serpentinite rocks underlying the springs. Many of these putative subsurface organisms may have been plucked from dense biofilm communities attached to serpentinites and then flushed out to the surface by the spring water. Residence times of ∼700 years have been estimated for spring water in this system (Cipolli et al., 2004), suggesting that the subsurface communities inferred by this study may persist for long time periods in isolation from the surface. We anticipate that the metagenomic inventories and data analysis tools described here will provide a foundation for future studies to investigate how these organisms make a living in such unusual conditions and to test whether such ecosystems can be supported solely by subsurface serpentinization-associated reactions.

Supplemental Information

File S1 16S rRNA taxonomy table, bacterial family

Click here for additional data file.

File S2 16S rRNA taxonomy table, bacterial sequence

Click here for additional data file.

File S3 16S rRNA taxonomy table, archaeal family

Click here for additional data file.

File S4 16S rRNA taxonomy table, archaeal sequence

Click here for additional data file.

File S5 Metagenome PhyloSift taxonomy table

Click here for additional data file.

File S6 Metagenome PhyloSift taxonomy Krona graphs

Click here for additional data file.

File S7 Metagenome metabolic pathway abundances

Click here for additional data file.

File S8 Metagenome key protein abundances

Click here for additional data file.

We are grateful to have had access to the excellent facilities of the ETH Stable Isotope Lab directed by Stefano Bernasconi. Esther Schwarzenbach, Melitza Crespo-Medina, Bridget Nelson, and Elena Amador provided valuable assistance and fun conversations in the field. Hilary Morrison, Sharon Grim, Mitch Sogin, and Rick Colwell facilitated the early stages of this project through the Census of Deep Life.

Additional Information and Declarations

Competing Interests

Author Contributions

DNA Deposition

Data Availability

The authors declare there are no competing interests.

William J. Brazelton conceived and designed the experiments, performed the experiments, analyzed the data, wrote the paper, prepared figures and/or tables, reviewed drafts of the paper.

Christopher N. Thornton analyzed the data, wrote the paper, reviewed drafts of the paper.

Alex Hyer and August A. Longino analyzed the data, reviewed drafts of the paper.

Katrina I. Twing conceived and designed the experiments, performed the experiments, analyzed the data, wrote the paper, reviewed drafts of the paper.

Susan Q. Lang conceived and designed the experiments, performed the experiments, analyzed the data, contributed reagents/materials/analysis tools, wrote the paper, reviewed drafts of the paper.

Marvin D. Lilley conceived and designed the experiments, performed the experiments, analyzed the data, contributed reagents/materials/analysis tools, reviewed drafts of the paper.

Gretchen L. Früh-Green and Matthew O. Schrenk conceived and designed the experiments, contributed reagents/materials/analysis tools, wrote the paper, reviewed drafts of the paper.

The following information was supplied regarding the deposition of DNA sequences:

VAMPS database (http://vamps.mbl.edu) under the project code DCO_BRZ and sample code Serp_LIG.

MG-RAST under IDs 4545477.3, 4545478.3, 4545479.3, 4545480.3, 4537863.3, 4537864.3, 4537868.3, and 4537869.3.

SRA identifier SRP049438 and BioProject PRJNA265986.

The following information was supplied regarding data availability:

https://github.com/Brazelton-Lab.

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
