# Peer review of "Metagenomic identification of active methanogens and methanotrophs in serpentinite springs of the Voltri Massif, Italy"

_PeerJ, doi:10.7717/peerj.2945_

## Round 0.1 · original submission · Major Revisions

Please address all comments related to replication of your analyses. We encourage you to provide additional data and/or modify your discussions to outline the limitation of the study.

Reviewer 1 ·

Basic reporting

see below

Experimental design

see below

Validity of the findings

see below

Additional comments

Brazelton et al presented a study to identify key microbes in a highly basic serpentinite spring ecocystem. amplicon sequencing for bacterial and archaeal was carried out and shotgun metagenomic sequencing was conduced for selected samples. microcosm incubations and isotope labelling were used to probe for active biogeochemical processes in this highly unusual system.

The study is carefully described and the manuscript itself is well written. One major concern is that the manuscript is fairly lengthy and should be significantly shorted in order to improve readability (see below). I have the following comments, hopefully to improve the presentation of the paper.

Generally speaking, I quite like the amplicon data presented. The depth of coverage is sufficient and the data presented in supple tables and figure 2 are well thought through. It would be even better if the authors have presented direct comparison of the amplicon data of the biological replicates.

The incubation experiments and isotope labelling are qualitative and I would have liked to see more quantitative data. for example, how fast is methane consumed aerobically and anaerobically? Do alternatively electron acceptors affect anaerobic methane oxidation?

My major concern comes from the metagenome analyses. the depth of coverage for each individual sample is rather poor, which is evident in data presented in table 3. Amplicon data demonstrated high inter-sample variability, therefore it is difficult to justify the combined assemble of all spring samples. Hence all subsequent analyses are biologically flawed. One way to improve the manuscript is therefore to significantly shorten this section.

minor comments
mcrA please use italic throughout the manuscript. (indeed use itablic for all gene names, eg. pmoCAB throughout)
please change 16S rDNA to 16S rRNA gene

Reviewer 2 ·

Basic reporting

This is a well-written manuscript that appears to have been thoroughly and carefully analyzed. I have no major criticism.

Experimental design

Generally a well-designed experiment. Little or no replication of some analyses (amplicon sequencing or metagenomics) resulting from low yields of samples. There is nothing that can be done about this now, nevertheless, I believe the results are still valuable and this should not preclude publication.

Validity of the findings

No comments

Additional comments

Congratulations on a nice study, which was informative and a pleasure to read.

I do have some comments that i would like considered.

1. I would like to know the closest relatives of the major OTUs identified in the study. Either a phylogenetic tree or even a table would suffice to indicate an accession number of a 16S reference sequence.
2. Please indicate what is known about the residence/transit time of the spring water. What can we expect for an incubation time of the water/microbes?
3. There are low cell counts in the spring water, but what about the possibility of biofilms deep within the system? I know that this will amount to speculation, but I think it is highly likely that biofilms have formed along rock surfaces where the chemical and physiological properties will be different from the water. I think this could be discussed.
4. Line 308, filters may also have clogged with carbonates, so this might not be solely a reflection of cell abundance.
5. What is the difference in absolute abundance of methanogens between river and spring water? A high abundance of Thaumarchaeota in the river water will reduce the relative abundance of methanogens. Is it possible that absolute abundance remains constant?
6. The absence archaeal sequences of GOR34-spring1-2013b is surprising when 16S analysis indicated that they were 10% of bacteria. This doesn't instill much confidence in either the 16s or metagenome analysis.
7. Is there a significant amount of dinitrogen on the spring water that could be fixed by nitrification? What about concentrations of fixed nitrogen (NH3, NO3 etc)?
8. You found mmoX, but what about the other genes encoding sMMO (ie. mmoYBDZ? Was this complete? This would hint at the coverage and therefore likelihood that pmoA is indeed not encoded in the genome of this methanotroph. Again, I am curious to know the closest relative of the Methylococcaceae (see comment 1 above).
9. Line 556. I do not understand how BLASTP of pmoA could give a hit to mmoX. These sequences are completely unrelated apart from catalyzing the same chemical reaction. It makes me question whether the KEGG annotation was correct.
10. What is acetylene degradation in Table 4?
11. Please check minor formating throughout, e.g. italicise gene names (mcrA) and subscript numbers in chemical formulae (e.g. NaHCO3 or CO2)

---

## Round 0.2 · Major Revisions

Dear William,

Thank you for your submission to PeerJ.

Your manuscript have been reviewed and reviewers find your work of interest. However, we were unable to obtain a re-review from the more critical reviewed from the first round, and so sought a new reviewer to comment on the revision. As a result a number of additional comments have been raised.

I would like to invite you to revise your manuscript and respond to all the comments below.

Reviewer 3 ·

Basic reporting

Reporting is not clear and sometimes ambiguous (see comments to the authors). Literature/background are sufficient. Figures, tables are OK but contradict the claims. The authors try to fit the results to their hypothesis, but it does not work.

Experimental design

The research described is primary research and is within the scope of PeerJ. Research design is poor. The study site is difficult in terms of collecting sufficient amounts of biomass. However, this is a poor excuse to produce poor data. Typically, a single cell approach is used in environments in which isolation of bulk DNA is problematic. The question is defined, but not really answered. State of the art techniques and softwares were applied, but they failed for the reason of low DNA quantity and possibly quality. Methods described sufficiently.

Validity of the findings

Poor data were produced and thus these bring little novel insight into the questions posed. Conclusions are not supported by data. Too many speculations.

Additional comments

This paper is written in a convoluted language and contains multiple repetitions, thus making it unnecessarily lengthy. My main concern is that the statements made throughout the text range from true to partly true to completely untrue. Just in the Abstract, it is stated that this report provides metagenomic and experimental evidence for active methanogenesis and methanotrophy. While methane oxidation indeed was measured in some samples, methanogenesis was only measured in one single sample, and not the one used for metagenomics. Metagenomic analysis detected no methanogen sequences in one spring, and only 1% sequences in another spring, which could represent contaminating sequences from the river or elsewhere. Later in the abstract, it is stated that methanogens and methanotrophs were heavily enriched. Not true. Looking at the Phylosift diagrams, I hardly see either methanotrophs or methanogens. They are very minor populations. This occurs throughout the text.

Primary research questions, as is typical of molecular detection and metagenomic approaches, are who is there and what are they doing and how. A gap to be filled is rather well defined. However, I think the insufficient amount of biomass employed in the experiments is the cause of poor data obtained in both the amplicon or metagenomic sequencing experiments and also in activity measurements. I am not sure how this can be resolved. Could more biomass be obtained? Did the authors calculate how much DNA would be needed to obtain high quality sequence datasets? Did authors consider applying single cell genomics as is typical for environments with low cell counts?

Early in the introduction the authors outline the goals of determining specific organisms that consume specific products of serpentinization, and specifically the role of methanotrophs and methanogens. They also state that previous reports have not provided any quantitative measurements of environmental distribution of the methanogens. Here, I do not think the authors achieved their goals, including quantitative measurements. They sort of detect Serpentinomonas, but its genome remains unassembled and thus no metabolic reconstruction could be done. Did Serpentinomonas become Hydrogenophaga in the metagenome? However, Hydrogenophaga is also unassembled. As already mentioned, only 1% sequences were predicted to belong to a methanogen. Overall, there are huge discrepancies between amplicon sequencing, metagenomic sequencing and quantitative PCR. I think this is all caused by low DNA concentration, and I feel for the authors, working with low cell count samples is difficult. However, the fact remains that these datasets are of very poor quality. The genomic bins are simply atrocious! 42% complete with 12% contamination representing two classes of Proteobacteria? This is not a bin, this is garbage. Another sign of poor data is the lack of matches in the databases, likely caused by chimeric sequences (also known as garbage). Transporters and stress response functions are not pathways, and likely you over-detect them because they are more easily recognized by automated tools. As to the real pathways, how can methanogenesis and methanotrophy pathways (note that only methane oxidation genes are detected, not methane oxidation and assimilation pathways!) be frequently detected when the abundances of both in the metagenomes is very low? I can see how the authors are trying but unable to reconcile their data, evident from the 10-page discussion and statements such as ‘…remains only incompletely confirmed…’ (L. 693).

Lots of literature is cited to support the findings. It appears that the authors decided a priori to find methanogenesis and methanotrophy, based on what has been found in marine environments. However, the small number of reads and respective identified methanogenesis enzymes would be entirely consistent with these organisms being anaerobic methane oxidizers instead, or being contaminants from river or other environments.

The figures also do not well support the major points of this paper. Sometimes they contradict the statements, for example, again, Phylosift files do not support abundant presence of Methylococcales or Methanobacteriales or even Comamonadales.

To conclude, I do not think the lead hypothesis is well supported by the results.

---

## Round 0.3 · accepted · Accept

Dear William,

Thank you for your submission to PeerJ.

I am writing to inform you that your manuscript - Metagenomic identification of active methanogens and methanotrophs in serpentinite springs of the Voltri Massif, Italy - has been Accepted for publication. Congratulations!